# High Transmission Potential of West Nile Virus Lineage 1 for *Cx. pipiens* s.l. of Iran

**DOI:** 10.3390/v12040397

**Published:** 2020-04-03

**Authors:** Hasan Bakhshi, Laurence Mousson, Marie Vazeille, Sedigheh Zakeri, Abbasali Raz, Xavier de Lamballerie, Navid Dinparast-Djadid, Anna-Bella Failloux

**Affiliations:** 1Malaria and Vector Research Group, Biotechnology Research Center, Pasteur Institute of Iran, Tehran P.O. Box 1316943551, Iran; hbakhshi89@gmail.com (H.B.); zakeris@yahoo.com (S.Z.); raz.biotech@gmail.com (A.R.); 2Institut Pasteur, Arboviruses and Insect Vectors, 75724 Paris, France; laurence.mousson@pasteur.fr (L.M.); marie.vazeille@pasteur.fr (M.V.); 3Unité des Virus Emergents (UVE), Aix Marseille Université, IRD 190, INSERM 1207, IHU Méditerranée Infection, 13005 Marseille, France; xavier.de-lamballerie@univ-amu.fr

**Keywords:** West Nile virus, vector competence, experimental infections, Iran

## Abstract

Vector competence is an important parameter in evaluating whether a species plays a role in transmission of an arbovirus. Although the protocols are similar, interpretation of results is unique given the specific interactions that exist between a mosquito population and a viral genotype. Here, we assessed the infection (IR), dissemination (DR), and transmission (TR) rates of *Cx. pipiens* s.l., collected from Iran, for West Nile virus (WNV) lineage 1a. We showed that *Cx. pipiens* s.l. mosquitoes in Iran were susceptible to WNV with IR up to 89.7%, 93.6%, and 83.9% at 7, 14, and 21 days post-infection (dpi) respectively. In addition, DR and TR reached respectively 92.3% and 75.0% at 21 dpi, and the number of viral particles delivered with saliva reached up to 1.33 × 10^5^ particles. Therefore, an unexpected high risk of WNV dissemination in the region where *Cx. pipiens* s.l. mosquitoes are well established should be considered carefully and surveillance measures implemented accordingly.

## 1. Introduction

Mosquito-borne viruses, such as dengue, chikungunya, Zika, and West Nile virus, are responsible for millions of disease cases and thousands of deaths each year [1]. West Nile virus (WNV; family *Flaviviridae*, genus *Flavivirus*) is the most widespread arbovirus in the world and can be fatal for humans [2]. WNV was isolated for the first time in 1937 from Uganda [3] and *Cx. pipiens* is considered as one of its primary enzootic vectors [4]. This vector is widely distributed in many countries [5]. WNV is now spread in Southern Asia, Northern Australia, Africa, and temperate regions of Europe [6,7]. The increasing traffic of goods and animals between countries offers multiple opportunities for the introduction of arboviruses in Iran, including WNV [8,9]. WNV is the most prevalent *Culex*-transmitted virus frequently reported in the country [10,11]. Approximately, 20% of WNV-infected people show a symptomatic infection [12]. Clinical symptoms are non-specific to the disease, including anorexia, fever, and headache. This virus is separated into two main lineages: (i) Lineage 1 is widely distributed and highly invasive and includes most strains responsible for outbreaks in the Middle East, India, Europe, and Australia [13]; this lineage is differentiated into three clades 1a, 1b, and 1c, with viruses of clade 1a found worldwide [14] and (ii) Lineage 2 was used to be endemic to some regions of Africa [14]; however, its circulation has been recently reported in Europe, Sub-Saharan Africa, Madagascar, and the Middle East, including Iran [7,15,16,17,18,19]. Lineage 2 has caused many human cases of neuroinvasive disease in Greece [20,21] and Italy [22], suggesting that it also can be as pathogenic as lineage 1. The amino acid identity among members of the two lineages is around 93.2%–94% [23].

WNV infection was first investigated in Iran in 1970 and, subsequently, additional investigations on mosquitoes, including *Cx. pipiens* s.l. as the main WNV vector [24], humans, horses, and birds, confirmed the presence of the virus in the country [25]. There is much serological and molecular evidence of WNV circulation in vectors, animal reservoirs, and dead end hosts in Iran [9,17,18,26,27,28,29,30]. Nevertheless, only few studies have succeeded in identifying the WNV lineages circulating in the country [15,17,18]: *Cx. pipiens* s.l. mosquitoes have been found to be infected with WNV belonging to lineage 2 [18]. In addition, lineage 1 of WNV has been recently detected in blood donors of Pakistan (east of Iran) [31] and in mosquito pools, *Aedes albopictus* and *Cx. pipiens* s.l., in Turkey (northwest of Iran) [32]. Other similar investigations have also proved the circulation of WNV lineage 1 in Turkey, bordered with Iran [33,34,35].

To get a precise view on how a vector-borne disease can become a global disease, information on vector competence of mosquitoes from different regions in the world is essential. Vector competence is an important parameter in evaluating whether a species plays a role in transmission of a pathogen in the field. Although assessments are based on common protocols, the outcome is unique, as different mosquito genotypes and virus genotypes (described under GxG interactions) are used [36].

Detection of an arbovirus in a vector does not indicate alone that the mosquito is a competent vector for the virus. The distribution of WNV in the country is widespread as it can be transmitted by a broad range of vectors and can infect different vertebrate hosts [25]. Despite the important role of vectors in the WNV biological cycle, there is no study on vector competence for WNV of mosquitoes in the region. More knowledge about transmission potential of mosquito vectors will provide important clues on risks of emergence or re-emergence of WNV in the area. Thus, for a better understanding of the mechanisms behind the maintenance and transmission of WNV, the present study was designed to evaluate the vector competence of *Cx. pipiens* s.l., the principal vector of WNV [37], and one of the most prevalent species reported [38] for the highly invasive and recently reported WNV lineage 1 in the region [31,32], based on measuring the infection, dissemination, and transmission rates of *Cx. pipiens* s.l. for WNV lineage 1a.

## 2. Materials and Methods

### 2.1. Ethic Statements

Laboratory rabbits were housed in the Institut Pasteur animal facilities accredited by the French Ministry of Agriculture to use their blood. Work on animals was performed in compliance with French and European regulations on care and protection of laboratory animals (EC Directive 2010/63, French Law 2013-118, 6 February 2013). All experiments were approved by the Ethics Committee #89 and registered under the reference APAFIS#6573-201606l412077987 v2.

### 2.2. Culex Pipiens Collection and Rearing

According to the standard dipping technique (350 mL dipper), Culicinae mosquito larvae were collected from Ghorogh forest (36°50′N 54°26′E), located in Golestan province, northeast of Iran, in 2015. Larvae of *Cx. pipiens* from *pipiens* (as well as the emerged adults) were identified using morphological characteristics [39] and molecular markers (Genbank accession number for cytochrome oxidase I gene: KY646203) and reared in insectaries under standardized conditions (26 ± 2 °C, 70 ± 10% relative humidity, 12 h:12 h light:dark photoperiod). Larvae were placed in dechlorinated water, supplemented with fish-powdered food. Emerging adults were collected daily and transferred into cages, where they were fed with a 10% glucose solution.

### 2.3. Vector Competence Assay

WNV experimental infections were run at the Institut Pasteur in Paris [40]. In brief, 5 to 7-day-old females were transferred to boxes, starved for 48 h in a biosafety level 3 insectary, and then allowed to feed for 20 min through a pig intestine membrane covering the base of a capsule of the feeding system Hemotek^®^, containing the blood-virus mixture maintained at 37 °C [41]. The infectious meal was composed of WNV suspension diluted in phosphate buffered saline (PBS)-washed erythrocytes from rabbit [40] to obtain a titer of 10^7.3^ pfu/mL. The adenosine triphosphate (ATP) was added as a phagostimulant at a final concentration of 5 × 10^−3^ M. The WNV belonged to lineage 1a (Genbank accession number: AY268132), which was isolated from a horse in France (Camargue) in 2000 [42]. Viral stocks were produced on *Aedes albopictus* cells C6/36 [43] after four passages on Vero cells and stored at −80 °C in aliquots until use. After exposure to the infectious blood meal, fully engorged females were transferred in cardboard containers and maintained with 10% sucrose at 28 ± 1 °C for 21 days. To evaluate the vector competence of *Cx. pipiens*, the infection rates (IRs) were assessed at 7 (29 samples), 14 (31 samples), and 21 (31 samples) days post-infection (dpi) by the RT-qPCR method. To confirm the infection of mosquitoes to WNV, three engorged mosquitoes were homogenized, and the presence of virus was confirmed by the RT-qPCR method at 0 dpi. The dissemination rate (DR), transmission rate (TR), dissemination efficiency (DE), and transmission efficiency (TE) of the samples at 21 dpi were assessed, as described below.

The total RNA of the 125 (0 dpi: 3; 7 dpi: 29; 14 dpi: 31; 21 dpi for IR assay: 31; 21 dpi for DR assay: 31) infected mosquitoes were extracted using a Macherey–Nagel NucleoSpin^®^ RNA extraction kit (Hoerdt, France). Total WNV RNA was quantified by RT-qPCR using the Bio-Rad CFX96™ Real-Time PCR Detection System and Power SYBR^®^ Green RNA-to-C_T_ 1-Step kit. For each reaction, we used 10.55 µL of distilled water, 12.5 µL of buffer 2X, 0.375 µL (10mM) of each designed forward and reverse primers (WN175up: 5′-GTGTTGGCTCTCTTGGCGTT-3′ and WN259low: 5′-AGGTGTTTCATCGCTGTTTG-3′), 0.2 µL of mix-enzyme 125X, and 1 µL of RNA. The reverse transcription was performed at 48 °C for 30 min. The qPCR conditions were 95 °C for 10 min, followed by 40 amplification cycles of 95 °C for 15 s and 60 °C for 1 min. Additionally, melt curve assay (65 °C to 92 °C: Increment of 0.5 °C 0:05) was carried out. For each run, the number of WNV RNA copies was calculated by absolute quantitation using a standard curve, which was generated using duplicates from 10^2^ to 10^8^ copies of RNA synthetic transcripts per reaction. Quantification of viral RNA was achieved by comparing the threshold cycle (Ct) values of samples to those of standards, according to the ΔC_t_ analysis.

To estimate viral transmission, saliva was collected from individual mosquitoes as described before [44]. For collection, the wings and legs were removed from each mosquito and the proboscis was inserted into a 20 µL tip, containing 5 µL of Fetal Bovine Serum (FBS). After 45 min of salivation, FBS containing saliva was expelled into 45 µL of Leibovitz L15 medium for titration. Infectious viral particles were detected in saliva by a plaque forming unit (pfu) assay on Vero E6 cell monolayers, as previously described [45]. Four parameters were calculated: (1) IR, corresponding to the proportion of *Cx. pipiens* species with an infected midgut among tested mosquitoes; (2) DR, indicating the proportion of *Cx. pipiens* species with an infected head among mosquitoes whose midguts were infected; (3) TR, defined as the proportion of mosquitoes with infectious saliva among *Cx. pipiens* species with an infected head, and (4) TE, showing the proportion of *Cx. pipiens* species with infectious saliva among all individuals tested [46].

### 2.4. Statistical Analysis

Rates, means, standard error (SE), and standard deviation (SD) were calculated, and statistical analyses were performed using the Stata software. *p*-values of <0.05 were considered statistically significant.

## 3. Results

### 3.1. Artificial Feeding

In total, 367 female adults were exposed to an infectious blood meal. Out of 367, 275 (74.9%) females successfully fed on the WNV-infected blood. Out of 275 blood fed females, 125 mosquitoes survived until the day of examination.

### 3.2. Viral Infection Over Time

Viral IRs at 7, 14, and 21 dpi were 26/29 (mean (%) ± SE: 89.7% ± 5.7), 29/31 (93.5% ± 4.5), and 26/31 (83.9% ± 6.7), respectively (Figure 1A); despite variations, IRs were not significantly different (χ^2^ test: *p*= 0.472). In addition, the viral load in bodies were 5.8 Log_10_ ± 0.9 (26) (mean ± SD (*n*)) at 7 dpi, 6.2 Log_10_ ± 1.7 (29) at 14 dpi and 6.8 Log_10_ ± 1.8 (26), showing a significant increase in viral loads over time (Kruskal-Wallis test: *p* = 0.0001) (Figure 1B).

### 3.3. Dissemination and Transmission at 21 Days Post-Infection

Among 31 mosquitoes examined at 21 dpi, 26 (83.9% ± 6.7) had an infected body and, among these, 24 (92.3% ± 5.3) had ensured viral dissemination with virus detected in the head. Among these 24 mosquitoes, 18 (75.0% ± 9.0) were able to transmit the virus with viral particles detected in mosquito saliva (Figure 2A). When examining viral RNAs, the number in bodies (6.8 Log_10_ ± 1.8) and heads (7.3 Log_10_ ± 1.0) were not significantly different (Mann–Whitney test: *p* = 0.56) (Figure 2B). When considering transmitted viruses, mosquito saliva contained a mean of 1.7 Log_10_ viral particles (± 1.1) in 18 mosquitoes, a number significantly lower than in the body and head (Kruskal–Wallis test: *p* = 0.0001) (Figure 2C). To measure the ability of mosquitoes to allow the virus to be transmitted with the saliva, the transmission efficiency (TE; 58.0 ± 9.0) was calculated, corresponding to the product of DE (77.4% ± 7.6) and TR (75% ± 9.0).

## 4. Discussion

To date, many experimental studies have focused on assessing the vector competence to evaluate the potential for WNV emergence and spread among mosquito populations [37,41,47,48,49,50,51,52,53,54,55], as well as other suspected hematophagous arthropods, including ticks [56]. Given its high local abundance and its feeding behavior towards birds and mammals, *Cx. pipiens* s.l. is a key vector of WNV in Iran [57] and in the world [58].

Recent investigations have detected field-collected *Cx. pipiens* mosquitoes, infected with WNV in Iran [9,18]. However, despite many other signs of WNV circulation based on serological and molecular detections in humans, horses, birds, and mosquitoes, the vector competence of this species remains unknown in the country. A large range of *Aedes* and *Culiseta* mosquito species (*Ae. caspius* mosquitoes) have been found to be infected with WNV in the northwest of Iran [15]), which are also efficient for transmitting WNV at even lower rates [51]. Interestingly, the experimental transmission of WNV by ticks [59] and by mites [60] has been attempted without any clear evidence. *Cx. pipiens* s.l. is described under two biological forms, *pipiens* and *molestus*, which are morphologically indistinguishable [61]. The Ghorogh Forest mosquito strain used in this study belongs to the *pipiens* form, which has been shown to be a primary vector of WNV in the Mediterranean basin [47].

After feeding on a viremic host, WNV must penetrate into the midgut epithelial cells and replicate. Subsequently, WNV must disseminate within tissues of the mosquito internal organs. To be injected into a new vertebrate host, WNV should infect the salivary glands [62]. The efficiency of barriers in mosquito vectors determines the level of mosquito vector competence [47]. Vector competence is mainly influenced by environmental temperature, as well as viral dose [63]. Arboviruses transmitted by mosquito vectors have seen their range of geographic distribution changing as a consequence of environmental modifications, creating suitable conditions to the establishment of their vectors combined with human activities, favoring passive transportation of mosquitoes. WNV presents one of the highest potentials to re-emerge as numerous mosquitoes, and birds are efficient transmitters with recurrent spillovers to equines and humans. Clear associations have been found between warm conditions and WNV outbreaks in various countries of the world [64]. Laboratory experiments have revealed that up to 100% of *Cx. tarsalis* mosquitoes become infected after feeding on blood containing a WNV concentration of 10^7.1^ pfu/mL, while in a blood containing 10^4.9^ pfu/mL, only up to 36% become infected [51]. It has been shown that the infectious doses required for infection of *Cx. pipiens* must be greater than 10^5.0^ pfu/mL [65]. In this investigation, we used a viral titer of 10^7.3^ pfu/mL and an incubation temperature of 28 °C.

The results of the present investigation revealed that IRs were not significantly different and varied from 83.9% to 93.5%. (Figure 1A); however, the viral load in bodies varied from 5.8 Log_10_ to 6.8 Log_10_, showing a significant increase in viral loads over time (Figure 1B). We also observed a viral DR and TR of 92.3% with virus detected in the head, containing a mean of 7.3 Log_10_ ± 1.0 viral particles, and 75% with viral particles detected in mosquito saliva, containing a mean of 1.7 Log_10_ at 21 dpi, respectively (Figure 2A–C). A TE of 58% was also calculated. The number of viral particles delivered with saliva was up to 1.33 × 10^5^ particles (Figure 2A). Based on the evaluation of IR, DR, and TRs, *Cx. pipiens* s.l. populations collected from the north of Iran were highly susceptible to WNV lineage 1a, with infection rates higher than 89.7% and viral loads in bodies higher than 5.8 Log_10_ from 7 dpi. At 21 dpi, viral infection, dissemination, and transmission were not hampered, suggesting a minor role of the two anatomical barriers, midgut, and salivary glands in the migratory route of the virus until mosquito saliva. Mosquitoes can then excrete an average of 1.7 Log_10_ viral particles (Appendix A).

For the WNV, mosquito genotype has a significant role in specific interactions between virus genotypes and mosquito vectors and influences the outcome of transmission of WNV [66]. Our results are consistent with former investigations in Lebanon [41] and the Maghreb region [47], carried out with the same WNV strain used in this study, corroborating the high IRs of *Cx. pipiens* colonies for WNV lineage 1. In Lebanon, Zakhia et al., (2018) showed IRs ranging from 94.7 to 100% from 3 to 19 dpi, DEs (compared to DR, DE indicates the proportion of mosquitoes with an infected head among all mosquitoes tested) increasing from 31.6% (3 dpi) to 94.7% (19 dpi), TEs gradually increasing from 10.5% at 3 dpi to 68.4% at 19 dpi, and the viral load reaching 1028 (± 405) pfu/saliva at 19 dpi [41]. Amraoui et al., (2012) also showed that *Cx. pipiens* mosquitoes from the Maghreb region were efficient experimental vectors to transmit WNV: at 21 dpi, TRs reached 80%, with a mean number of infectious particles in saliva of 1.7 ± 0.9 log_10_ pfu, DRs ranged from 59.1% to 100%, and TRs varied from 25% to 83.3% [47]. On the other hand, it has been shown that *Cx. pipiens* mosquitoes were moderately efficient vectors of this virus in France with a TR of 15.8% at 14 dpi [48], as did *Cx. tarsalis* [54], *Cx. modestus* [49], *Cx. Quinquefasciatus*, and *Cx. nigripalpus* [52].

Vector competence assays should be combined with an active monitoring of WNV lineage 1 circulation in Iran [31,32]. We used a lineage 1 strain to assess vector competence of *Cx. pipiens* s.l. mosquitoes; we suggest that vector competence would be quite similar if testing with other strains of lineage 1. Further studies should be done using strains of lineage 2 [15,17,18], which can be as pathogenic as lineage 1.

In conclusion, our results showed an unexpected high transmission potential of WNV lineage 1 by *Cx. pipiens* s.l. of Iran, meaning that outbreaks are to be feared as active circulation of WNV operates in the region. Thus, local health authorities are suggested to establish an active surveillance with an early detection of WNV infection in field-collected mosquito vectors.

## Figures and Tables

**Figure 1 viruses-12-00397-f001:**
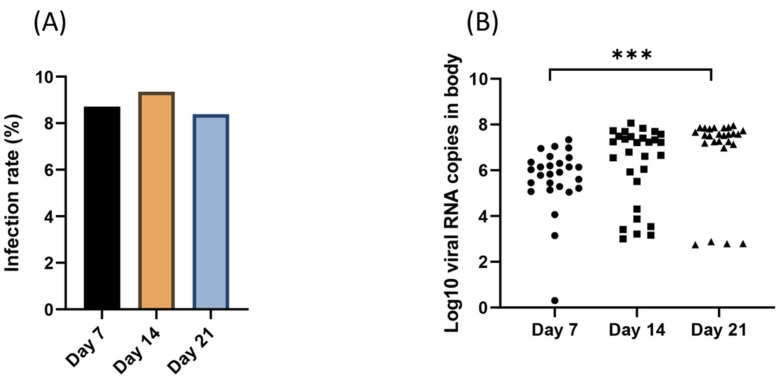
Infection rate (**A**) and viral RNA copies measured in bodies (**B**) at 7, 14, and 21 days after exposure of *Culex pipiens* s.l. to West Nile virus at a titer of 10^7.3^ pfu/mL. Viral RNA particles were detected within bodies by RT-qPCR. ****p* ≤ 0.001.

**Figure 2 viruses-12-00397-f002:**
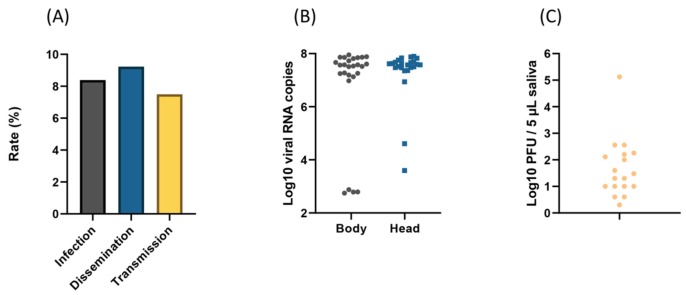
Infection, dissemination, and transmission rates (**A**); viral RNA copies in the body and head (**B**); and viral particles in saliva (**C**), 21 days after challenging *Culex pipiens* s.l. with West Nile virus at a titer of 10^7.3^ pfu/mL.

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
