# Peer review of "High Transmission Potential of West Nile Virus Lineage 1 for Cx. pipiens s.l. of Iran"

_viruses, 2020, doi:10.3390/v12040397_

Round 1

Reviewer 1 Report

After reviewing the corrected version of the manuscript and the answers of the authors to the first review, it can be considered that most of the questions have been corrected in this version. Nevertheless, the most relevant question has not been satisfactorily answered.

In the first revision it is proposed that the best lineage to carry out the experiment is lineage 2, because it is the lineage that have been detected in the country. The authors explain that in recent years lineage 1 has been detected in the neighbouring countries Pakistan (2016-2018) and Turkey (2016-2017). These data are more recent that the detection of lineage 2 in Iran (2015). It is a correct argument to use lineage 1 in the experiment. Nevertheless, the other reason given in the authors answer is not satisfactory. In the first sentence of the answers document you write: “It has been suggested that viral isolates of lineage 1 are associated with clinical human encephalitis while lineage 2 isolates are maintained in enzootic foci (Peterson and Roehrig 2001)”. This statement is not currently true. In fact in Europe lineage 2 have caused in recent years a high number of human encephalitis cases, causing the most relevant outbreaks caused by West Nile virus in the continent in countries as Italy and Greece and in some cases (Italy) has replaced to lineage 1 (Viruses 2019, 11, 814; doi:10.3390/v11090814), (Archives of Virology (2019) 164:1673–1675 https://doi.org/10.1007/s00705-019-04243-8).

It is required that the authors revise the recent literature to analyse correctly the incidence of the 2 main WNV lineage. Considering that lineage 2 can be as pathogenic as lineage 1 and strains detected in Pakistan and Turkey has not showed to be highly pathogenic, the only reason that can be used to explain the use of a lineage 1 strain is that the detection in the area is more recent. Consequently, you should explain in the article that the conclusions of the experiment could be extrapolated to lineage 1 strains (and considering that differences between strains of the same lineages can occur). You should consider include in the discussion the limitations of the assay, because lineage 2 strains (that has been circulating in Iran and can be as pathogenic as lineage 1 in some areas) has not been assayed.

Please, after the necessary bibliographic review, consider change the sentences in relation to lineage 2 relevance and pathogenicity in different sections of the manuscript (page 3, lines 50-51; page 4, lines 66-67; page 11, lines 216-220).

Other minor issues:

A more detailed discussion comparing your results with results obtained in Lebanon and Maghreb region (references 36 and 42) can be included. You can detail that the WNV strain used is the same.

Figures 1A and 2A. Change numbers in vertical axis (infection rate (%)), being a percentage, you should change “0.0, 0.2, …..1.0” to “0, 2, …..10”

Figure 1B. Indicate in figure legend the meaning of “***”

Author Response

Answers to reviewer #1:

Point #1: In the first revision it is proposed that the best lineage to carry out the experiment is lineage 2, because it is the lineage that have been detected in the country. The authors explain that in recent years lineage 1 has been detected in the neighbouring countries Pakistan (2016-2018) and Turkey (2016-2017). These data are more recent that the detection of lineage 2 in Iran (2015). It is a correct argument to use lineage 1 in the experiment. Nevertheless, the other reason given in the authors answer is not satisfactory. In the first sentence of the answers document you write: “It has been suggested that viral isolates of lineage 1 are associated with clinical human encephalitis while lineage 2 isolates are maintained in enzootic foci (Peterson and Roehrig 2001)”. This statement is not currently true. In fact in Europe lineage 2 have caused in recent years a high number of human encephalitis cases, causing the most relevant outbreaks caused by West Nile virus in the continent in countries as Italy and Greece and in some cases (Italy) has replaced to lineage 1 (Viruses 2019, 11, 814; doi:10.3390/v11090814), (Archives of Virology (2019) 164:1673–1675 https://doi.org/10.1007/s00705-019-04243-8). It is required that the authors revise the recent literature to analyse correctly the incidence of the 2 main WNV lineage. Considering that lineage 2 can be as pathogenic as lineage 1 and strains detected in Pakistan and Turkey has not showed to be highly pathogenic, the only reason that can be used to explain the use of a lineage 1 strain is that the detection in the area is more recent. Consequently, you should explain in the article that the conclusions of the experiment could be extrapolated to lineage 1 strains (and considering that differences between strains of the same lineages can occur). You should consider include in the discussion the limitations of the assay, because lineage 2 strains (that has been circulating in Iran and can be as pathogenic as lineage 1 in some areas) has not been assayed. Please, after the necessary bibliographic review, consider change the sentences in relation to lineage 2 relevance and pathogenicity in different sections of the manuscript (page 3, lines 50-51; page 4, lines 66-67; page 11, lines 216-220).

We thank the reviewer 1 for his/her valuable comments. We deleted the sentence "In brief, isolates of lineage 1 are associated with clinical human encephalitis while lineage 2 isolates are maintained in enzootic foci", added your references regarding the presence of WNV lineage 2 in Greece and Italy (Line 54), and revised the sentences regarding the incidence of the two lineages, considering that the lineage 2 can be as pathogenic as lineage 1 (Line 53-55).We stated that we used lineage 1 based on the more recent report of the presence of this lineage in the region (Lines 84, 27-278). We also considered the limitations of our study (Lines 272-276). Please see the highlighted sections in Introduction and Discussion:

  • Lines 47-59: "This virus is separated into two main lineages: (i) lineage 1 is widely distributed and highly invasive, and includes most strains responsible for outbreaks in the Middle East, India, Europe, and Australia [13]; this lineage is differentiated into three clades 1a, 1b, and 1c with viruses of clade 1a found worldwide [14] and (ii) lineage 2 was used to be endemic to some regions of Africa [14], however, its circulation has been recently reported in Europe, Sub-Saharan Africa, Madagascar, and the Middle East including Iran [7, 15-19]. The lineage 2 has caused many human cases of neuroinvasive disease in Greece [20, 21] and Italy [22], suggesting that it also can be as pathogenic as lineage 1. The amino acid identity among members of the two lineages is around 93.2–94% [23].

WNV infection was first investigated in Iran in 1970 and subsequently, additional investigations on mosquitoes including Cx. pipiens s.l. as the main WNV vector [24], humans, horses and birds confirmed the presence of the virus in the country [25]."

  • Lines 80-84: “Thus, for a better understanding of the mechanisms behind the maintenance and transmission of WNV, the present study was designed to evaluate the vector competence of pipiens s.l., the principal vector of WNV [37] and one of the most prevalent species reported [38], for the highly invasive and recently reported WNV lineage 1 in the region [31, 32], based on measuring the infection, dissemination and transmission rates of Cx. pipiens s.l. for WNV lineage 1a.”
  • Lines 272-276: "Vector competence assays should be combined with an active monitoring of WNV lineage 1 circulation in Iran [31, 32]. We used a lineage 1 strain to assess vector competence of pipiens s.l. mosquitoes; we suggest that vector competence would be quite similar if testing with other strains of lineage 1. Further studies should be done using strains of lineage 2 [15, 17, 18] which can be as pathogenic as lineage 1."

Reviewer 2 Report

The purpose of the study and what it adds to the scientific community is unclear. It is known that Culex mosquitos play a role in WNV transmission. Although the paper explains the methods and results, this manuscript appears to be based on only one small study and the data presented would be better presented as part of a larger story, especially one that reflects the relationship to human disease in the region.

Author Response

Point #1: The purpose of the study and what it adds to the scientific community is unclear. It is known that Culex mosquitos play a role in WNV transmission. Although the paper explains the methods and results, this manuscript appears to be based on only one small study and the data presented would be better presented as part of a larger story, especially one that reflects the relationship to human disease in the region.

We added several information in the text to justify the purpose of our study:

  • Lines 45-47: "Approximately, 20% of WNV-infected people show a symptomatic infection [12]. Clinical symptoms are non-specific to the disease, including anorexia, fever, and headache."
  • Lines 57-59: "WNV infection was first investigated in Iran in 1970 and subsequently, additional investigations on mosquitoes including pipiens s.l. as the main WNV vector [24], humans, horses and birds confirmed the presence of the virus in the country [25]."
  • Lines 74-80: "Detection of an arbovirus in a vector does not indicate alone that the mosquito is a competent vector for the virus. The distribution of WNV in the country is widespread as it can be transmitted by a broad range of vectors and infect different vertebrate hosts [25]. Despite the important role of vectors in WNV biological cycle, there is no study on vector competence for WNV of mosquitoes in the region. More knowledge about transmission potential of mosquito vectors will provide important clues on risks of emergence or re-emergence of WNV in the area. "
  • Lines 272-281: "Vector competence assays should be combined with an active monitoring of WNV lineage 1 circulation in Iran [31, 32]. We used a lineage 1 strain to assess vector competence of pipiens s.l. mosquitoes; we suggest that vector competence would be quite similar if testing with other strains of lineage 1. Further studies should be done using strains of lineage 2 [15, 17, 18] which can be as pathogenic as lineage 1.

In conclusion, our results showed an unexpected high transmission potential of WNV lineage 1 by Cx. pipiens s.l. of Iran, meaning that outbreaks are to be feared as active circulation of WNV operates in the region. Thus, local health authorities are suggested to establish an active surveillance with an early detection of WNV infection in field-collected mosquito vectors.

Reviewer 3 Report

The changes made in the revision improved the manuscript which gives relevant inputs on WNV transmisison.

Author Response

Point #1: The changes made in the revision improved the manuscript which gives relevant inputs on WNV transmisison.

We thank the reviewer 3 for his/her suggestions and comments which have improved the quality of our manuscript.

Round 2

Reviewer 1 Report

After a second revision of the manuscript and considering changes perfomed mainly in Introduction and discussion, I  consider that it now warrants publication in viruses.

Minor issue: The  issue in relation to figure 1 exposed in first revision has not been solvedChange numbers in vertical axis (infection rate (%)), being a percentage, you should change “0.0, 0.2, …..1.0” to “0, 2, …..10”  If you don't change it in the graph you indicate that vira IRs at 7, 14 and 21 were 0.897%, 0,935% and 0.839% instead the real data (89.7%, 93.5% and 83.9%)

Author Response

Point #1: The issue in relation to figure 1 exposed in first revision has not been solved

Change numbers in vertical axis (infection rate (%)), being a percentage, you should change “0.0, 0.2, …..1.0” to “0, 2, …..10” If you don't change it in the graph you indicate that vira IRs at 7, 14 and 21 were 0.897%, 0,935% and 0.839% instead the real data (89.7%, 93.5% and 83.9%)

We changed the numbers in vertical axis of figures 1A and 2A to "0, 2, …..10".

This manuscript is a resubmission of an earlier submission. The following is a list of the peer review reports and author responses from that submission.

Round 1

Reviewer 1 Report

The manuscript "High transmission potential of West Nile virus lineage 1 for Cx pipiens s.l. of Iran" by Bakhski et al., reports information about vector competence of local mosquitoes populations from Iran for WNF. This article reveals a high competence, and could indicate a high risk of transmission by this mosquito species in the area.

However, the manuscript has some deficiencies.
1.- First of all, the viral strain used for the study is a lineage 1 strain. Nevertheless the strains that have been detected in Iran belong to lineage 2 (reference 13 and Shah-Hosseini et al.  Emerging Infectious Diseases (2014), 20(8): 1419-1421. Consequently the best option to analyse competence index for the evaluation of the risk in Iran is a lineage 2 strain (as similar as possible to Irani strains). In fact in the introduction you have included the sentence: “..the interpretation of the results in unique given the specific interactions that exist between a mosquito population and a viral genotype”. You should justify why have you used a lineage 1 strain and consider the limitations of the use of other lineage in the conclusions regarding the risk of transmission in Iran where other lineage seems to be present.

2.- Introduction is very shallow. If it is possible it should include some information about WNV circulation in Iran. when and where samples were taken.

3.- More details are necessary in Material and Methods. In relation to viral strain used you should include information about passage history. As explained in reference 10 passage number and cell type used have a high influence in vector competence studies. You should also include the GenBank reference of viral strain used.

In relation to viral detection by RT-qPCR you didn´t use one of the diverse methods used by other authors to detect WNV. Consequently you should include details in relation of the way you performed the standard curve for quantitation.

4.- Discussion is also shallow. Different works have been published about vector competence of Culex pipiens in neighbouring countries and in Maghreb Region with the participation of some of the authors of these work. I suggest a comparison with the results obtained in mosquitoes from other countries.

4.- Only one experiment has been done. The replication of the experiment with other pool of mosquitoes will highly improve the work.

Minor comments:

Page 1, lines 14-15: change the sentence “Data on vector competence of Cx. Pipiens in Iran is insufficient; these data are essential to follow the expansion of West Nile virus (WNV), the most prevalent Culex transmitted virus reported in Iran.”. You don´t need data on vector competence to follow the expansion of West Nile, in fact serological studies are carried to follow situation of West Nile.

Page 3, lines 89-90, 99-100 and 107-108: You have performed only one experiment. What does mean the SE in IRs and DRs and TRs data? The application of a standard error and error bars in figures 1A and 2A is not required.

Figure 1B: change “Log10 viral RNA in body” to “Log10 viral RNA copies in body”

Figure 1B: There are 27 point at Day 7, but only 26 mosquitoes were positive for WNV (table S1). The value corresponding with the point with lower viral RNA is not present in table S1. Solve this issue.

Figure 2C: “Change Log10 virual particles in saliva” to “Log10 PFU/……saliva”. Include the volume of saliva corresponding with the PFU observed.

Reviewer 2 Report

Overall and interesting study and necessary to the community. Comments by section:

Abstract: Well written, clear.

Introduction: lacks important background and set up. Very broad statements that do not set up the reader for what to expect and why this study is needed.

Methods:

Line 81: “StataCorp LP, Texas, and USA). And should be removed.

Results: clear, but very brief

Discussion: fairly clear, would be nice to indicate how these results can be applied and what may come next

Reviewer 3 Report

The present experimental study on assessing the vector competence of Cx. pipiens s.l. mosquitoes collected in Iran is well described and results are clear. However, the authors investigate only one species of Culex, already known to harbor WNV in Iran. To add relevance to this report, authors should better frame it in the existing literature panorama, in Iran and in other neighbouring countries.

Authors should discuss more extensively (both in introduction and discussion) the information already present in literature about WNV, Culex and Iran and underline that the results are in agreement with those obtained from seroprevalence studies in captured mosquitoes in Iran; an example of an additional reference which should be cited is: Bagheri et al. 2015. Vector Borne Zoonotic Dis. West Nile Virus in Mosquitoes of Iranian Wetlands.

In addition, they should better explain the influence of environmental conditions in vector competence and so the importance of assessing competence in different countries.